# Personal attitude or experience? Which factors influence residents' acceptance of mixed-income communities?

**Duc Trung Luu**, **Dao Chi Vo**, **Jeongseob Kim** *

Department of Urban and Environmental Engineering, Ulsan National Institute of Science and Technology, Ulsan, South Korea

* jskim14@unist.ac.kr

**Data Availability Statement:** All relevant data are within the manuscript and its Supporting Information files.

**Funding:** This work was supported by the National Research Foundation of Korea (NRF) Grant funded

## Abstract

Although many researchers and policy makers have argued that social mixing could contribute to sustainable communities, most people still prefer to live in a homogeneous rather than a diverse community. Considering the large gap between the political need for social mixing and people's preference, it is essential to understand residents' perceptions and preferences regarding socially-mixed neighborhoods in order to promote sustainable community development. This study explorers residents' willingness to accept living in mixed-income communities in Korea, with attention to various levels of income mix. This study conducted an online survey of 2,000 respondents living in seven metropolitan cities in Korea, including Seoul. The study aimed to investigate residents' comfortability and willingness to move into different mixed-income communities. The results showed that residents with higher openness to diversity are more likely to accept mixed-income communities, but frequent interaction with low-income people reduces higher-income people's willingness to accept mixed-income communities. As both personal attitudes and experience are important determinants of individuals' social mix preference, a more systematic community development strategy is required to achieve successful social mixing.

## Introduction

Social mix has been considered an important and promising policy agenda in many countries to ensure sustainable, inclusive, and equitable communities [1]. Specifically, mixed-income development has been more prevalent in the United States and other Western countries as a means to tackle poverty concentration and social segregation caused by the historical practice of high-density public housing [2–4]. Social mix policy and planning aim to provide low-income residents with more occupational access in order to reduce income segregation and discrimination in a move toward social equity and universal well-being [5–8]. The strategy fundamentally targets declining and deprived neighborhoods with the goal of transforming them into more vibrant, accessible, and safer communities through "positive gentrification,"

by the Korean Government(MSIP) (No.NRF-
2015R1A5A7037825). The funders had no role in
study design, data collection and analysis, decision
to publish, or preparation of the manuscript.

**Competing interests:** The authors have declared
that no competing interests exist.

which is intended to capitalize on the available resources and attract higher-income residents for revitalization [3, 5].

On the other hand, there has been much criticism that in the actual field of community development, social mix policy and planning do not work like the theory from which they originate. From the neoliberalist perspective, there have been many concerns regarding profit generation in inner-city redevelopment projects [9], in which the unbalanced roles of different income classes could result in social tension, disorders, stigma, and social segregation [10–12]. Social mix projects, such as HOPE VI, are being criticized for overlooking potential inequality issues by emphasizing physical changes to public housing through design-based intervention [10]. Further, the advantages of social mix models could be limited because the process of social capital formation, such as cross-class interaction and networking, in socially-mixed communities is not actually effective. The Gautreaux program in Chicago and the Yonkers program in New York showed that incoming low-income residents often had a limited relationship with the higher-income residents in their destination neighborhood and thus were observed to strongly maintain their existing network in their former community [13–15]. Moreover, historical social housing could itself be a structural barrier to social mix adoption or a factor reducing governmental willingness to implement truly effective social mix policy due to public concern about the social deviance of social housing residents [12, 16]. Consequently, social mix policy is generally supported by politicians and frequently adopted as policy agenda, but it rarely solves residential segregation and inequality issues, particularly in cities [17].

Unlike the United States and other Western countries, where social mix policies comprehensively consider both race and income issues, social mix policy debates in Asian countries focus on a mix of different income groups or generations, with less racial diversity. For instance, the Korean government implemented a mixed-income and mixed-tenure housing program called Bogeumjari Housing during the Lee Myung-bak administration [18]. Through this program, permanent public housing targeting extremely low-income citizens, national public housing targeting low- and moderate-income groups, affordable housing for sale, and market rate housing for sale were simultaneously offered in the same residential complex in order to create housing stability and socially-mixed communities. More recently, the Korean government announced a plan to ensure social mix and prevent the stigmatization of public housing by integrating different types of public housing to serve different income classes into a unified type and improving the quality of the buildings [18]. However, social exclusion has not been fully tackled due to the social stigma that divides public and private housing occupants, causing structural barriers to effective social mix policy [19].

Explicitly, social mix varies by mixture type between public and market-driven housing or between different incomes through individual perception and political acceptance amongst income intervention groups regarding socially-mixed communities. Social mix can be achieved through the so-called "pepper-potting" approaching, which aims to increase exposure to positive role models among the disadvantaged in keeping with social control theory [20]; however, in some cases, social mix can also negatively disrupt social networks and local institutional support for the disadvantaged and subject them to the experience of stigmatization and inter-group conflict in socially-mixed communities [12, 20, 21]. Therefore, different levels of mix can provide different outcomes, as debated among scholars and planners [22]. Considering that positive perceptions and voluntary participation in social mix programs can promote the formation of social capital among different social groups, it is important to understand individuals' acceptance of different levels of mix.

In this regard, this study aims to explore residents' acceptance of different levels of income mix to identify the driving factors affecting their perceived attitude to social mix. We hypothesize that individuals' acceptance of socially-mixed communities varies not only by

demographic and socioeconomic characteristics but also by cognitive perception, which is formed through experiences of cross-class interaction in the context of different levels of mix. Specifically, this research addresses three research questions: (1) How do lower-income and higher-income people differ in their perception of mixed-income communities? (2) How does acceptance of mixed-income communities vary according to the level of income mix? (3) How do individuals' attitudes and experiences regarding different classes affect their acceptance of mixed-income communities?. To answer the research questions, we developed a survey questionnaire and administered it to 2,000 people in Korea's major cities in 2018. By considering multiple dimensions of residents' acceptance of mixed-income communities amongst different income groups, the study can provide a comprehensive view of social mix policy from the consumer perspective.

The paper is organized into four sections as follows. The next section provides theoretical and empirical reviews of the socially-mixed approach, focusing on residents' perception. The subsequent sections demonstrate the study area, data, and methods of analysis. The final two sections present results and findings, and concluding remarks with some policy suggestions.

## Literature review

### Theoretical foundation of social mix policy

Since the middle of the 19[th] century, socially-mixed communities have been a desirable town planning goal under the assumption that social interaction, social awareness, and a sense of community can be achieved through social mix [6]. The Bournville model village, founded by George Cadbury, is known as one of the first examples of a socially-mixed community, "explicitly enforcing (a) mix of renters and owner-occupiers, high-status and low-status Cadbury employees and also non-employees(p.5)" [6]. Since the model village, different types of socially-mixed communities have been suggested and implemented as part of housing and community development policy, evidenced by the mixed-tenure communities in the United Kingdom and the mixed-income communities established through the HOPE VI projects in the United States [23, 24].

Social mix has often targeted low-income residents with the aim of reducing social segregation and discrimination, but it also encourages aesthetic diversity and cultural cross-fertilization to strive for equality amongst different social classes [6]. Additionally, advocates of social mix policy believe that healthy and livable communities or revitalization opportunities can be gained through efforts to increase income diversity in neighborhoods [25]. Many researchers argue that a well-balanced social mix policy could support the formation of inclusive, safe, and accessible communities through "positive gentrification," while also preventing social exclusion and stabilizing the tax base [3, 5, 26, 27].

For these reasons, in recent decades, a resurgence of interest in mixed-income societies has occurred in many countries. In the United Kingdom, income-mixing strategies are being implemented to attract higher-income tenants to existing social housing stocks by improving the physical quality and management services of social housing [28]. In the United States, deteriorated and deprived public housing complexes are being redeveloped into mixed-income and mixed-tenure communities through the HOPE VI program [29]. As noted earlier, Korea also actively introduced social mix strategy to public housing development by co-locating public housing, affordable rental housing, and market-rate housing for sale through the Bogeum-jari Housing project [18]. The series of regeneration strategies related to housing and the environment in Australia, with attention to employment, residential participation, and inclusive neighborhoods, is also an example of mixed-income policy [12, 26].

In spite of its popularity as a policy agenda, social mix policy often fails to achieve its goal of creating socially-mixed communities. A growing body of literature posits that social exclusion has not been fully tackled due to the social stigma that divides the occupants of public and private housing [19]. This could have originated from many middle- and high-income people's negatively biased perception of the low-income group due to the heavy tax burden imposed by public housing programs [30]. As noted in the Introduction, much effort has been made to induce social mix in the housing market by mobilizing various policy incentives, but the desired effect of creating social ties and capital among different social groups is limited [10]. Considering the naturally higher prevalence of a preference for socioeconomic and cultural similarities [31], social mix may be a policy goal that is difficult to realize in the housing market, given that individuals can move freely. If the main social mix challenge lies in people's preferences and perceptions, it is necessary to gain a better understanding of individuals' attitudes toward social mix.

## Acceptability of social mix

Social mix policy performance could vary depending on individuals' perception and acceptance of various types and levels of mix between public and market-driven housing or between different incomes. Galster [20] proposed the "pepper-potting" approach, which emphasizes increasing exposure to positive role models among disadvantaged people, in keeping with social control theory. A more tolerant neighborhood can provide opportunities for low-income people to build social ties with members of other income groups within the community and lead to better educational performance among low-income children [6, 26]. However, the type and level of mix can trigger different perceived attitudes among members of the intervention group because people's tolerance of heterogeneous society differs [22]. Some literature has posited that social mix policy has faced different levels of resistance from different social classes, particularly the middle- and high- income groups. Blanc [32] argued that social mix strategy is not fully compatible with equality and freedom, since the right to choose whom to live with is a debatable issue. It could lead to social distance between the lower- and higher-income groups or even the rejection of affluence among the poor as seen in some stigmatized French neighborhoods [11], possibly resulting in stronger social resistance from certain social classes that exceeds the level of hostility involved in ethnic segmentation [32]. It could also disrupt social networks and local institutional support for the disadvantaged and subject them to destabilization and social inter-group conflict in socially-mixed communities [12, 20, 21]. This does not mean that the values of social mix strategy should be abandoned; however, it is important to rethink its adoption, particularly from the bottom up, with attention to residents' recognition and acceptability.

With regard to social class diversity, the resident viewpoint varies based on residents' political characteristics. From the lens of the qualitative approach, Rose [27] tried to explore four sub-groups, namely the "ignorant/indifferents," the "NIMBYies," the "tolerants," and the "egalitarians;" the latter three were found in all his studied neighborhoods (p. 278). In this study, a respectable reflection of social mix was found due to the possible affection for a cosmopolitan self-image. In the same vein, in contrast to conservative individuals who exhibit strong resistance to social mix, those who highly value openness to change have a higher intention to accept social diversity [33]. Local resistance to social mix cannot simply be considered NIMBY because it is the result of comprehensive reflection on the understanding of and attitudes toward the nature of social housing, related policy, and homeownership. As Ruming [34] noted, residents who support social housing policy in general could resist it in their neighborhood. In this case, the space for local resistance should be decreased through the provision

of clear, transparent intervention goals in alignment with the position of policy acceptance [34].

Political acceptance of social mix can vary not only by individual demographics but also by cognitive perception formed through cross-class interaction experience. Socioeconomic factors such as race, economic status, age, and education level can lay the foundation for residents' perception and acceptance of social mix [19, 35–37]. Housing tenure has been identified as a critical element in individual acceptance of social mix developments [30, 38]. Researchers have also found that residential satisfaction and length of stay in a neighborhood are also important determinants of the acceptability of social mix [35, 39]. Neo-liberal thoughts could influence residents' recognition of social mix, affecting the dynamics of their acceptance [27] The processes of economic reform and the institutional transition could affect residents' ideology and value judgment, leading to a change in residents' preferences to inclusive housing policies [35]. Specifically, compared to the elderly having a strong belief in social equality, the younger generation who is likely to support economic individualism, tends to have lower preferences for inclusionary housing in Jiangsu Province, China [35].

In addition, cross-class interaction is a key benefit of social mix in terms of reducing social isolation [6]. It has been found to positively affect individuals' personal growth and enhance the social network, improving social control [40, 41]; however, it still has a negative or unclear impact on different social classes, such as social tension, disorders, stigma, and segregation [10–12]. According to Pettigrew (1998)'s intergroup contact theory [42], the outcomes of social interactions between different groups are determined through the four processes: "learning about the outgroup, changed behaviors, affective ties, and ingroup reappraisal (p.65)". Based on this intergroup contact theory, increasing the frequency of interaction with social housing tenants is likely to lead to the acceptance of social mix and reduce stigma [12, 16], thus easing social conflicts and tension between public and market-driven housing tenants [19, 43]. However, interactive experience due to physical proximity to members of different income groups is not enough to change perceptions; it is necessary to take the level and quality of social interaction into account [12].

In sum, a growing body of literature has identified the factors affecting the acceptability of social mix. Acceptance can vary according to demographic elements, political intention, and social interaction. In line with existing studies, this study aims to explore the factors affecting people's willingness to accept mixed-income communities in the Korean context, with an emphasis on individuals' attitudes and experiences. This study comprehensively considers both attitudes toward and experiences of social mix to analyze people's preference regarding mixed-income communities in the Asian context with the overarching goal of filling the knowledge gaps in community development studies.

## Materials and methods

### Case: The Korean context

To support low-income households, the Korean government has implemented public housing policy since 1989. Like many countries worldwide, Korea's housing policy eventually faced issues of poverty concentration, resulting in social stigma and discrimination [18, 19]. To deconcentrate poverty in public housing developments, the social mix strategy has also been considered in Korea [18, 19]. In Western countries, socially-mixed communities mainly consist of racial and income mixing, but this study only focuses on the latter, since Korea is a racially homogenous country [44]. Mixed-income strategy in Korea originated from inclusionary housing on residential redevelopment sites. By law, all residential redevelopment projects have to build at least 17% of the units as affordable rental housing for displaced tenants [18].

City governments purchase these units, paying only the building cost, not the land cost; they then provide them to lower-income tenants. The initial debate over social mix on such redevelopment sites focused on the physical layout of the affordable rental units. Affordable rental units were often built separately from the market-rate housing units, that is, with separate entrances and playgrounds, resulting in discrimination against and stigmatization of low-income tenants. In response, regulations have been imposed prohibiting the separation of affordable and general housing units within a complex. Further, the social mix design is recommended, so that affordable rental housing and general housing cannot be physically distinguished in communities [19]. Recently, expanding income mix strategy so that it applies to all public housing projects sites has been considered to unify different types of public housing programs into a single type of program.

Although discussions about income mix in public housing are increasing in Korea, public housing NIMBYism still exists, and governments often face difficulties finding locations for public housing due to strong opposition from existing residents. Therefore, it is very important to gain a better understanding of people's acceptance of mixed-income communities in order to make the implementation of the mixed-income strategy in public housing successful.

## Survey and data

To explore residents' preference and acceptance of mixed-income communities, where low-income residents live together with higher-income residents, we conducted an online survey of 2,000 people. As this study examines people's willingness to accept mixed-income communities, the survey respondents are the general public rather than the residents living in mixed-income communities. This survey used stratified sampling based on population for each age group (20s, 30s, 40s, and 50+) in each city from the Population and Housing Census. The total number of respondents was 2,000 from the seven largest metropolitan cities in Korea, namely Seoul, Busan, Daegu, Gwangju, Daejeon, Incheon, and Ulsan. These are the only seven metropolitan municipalities in Korea and their population ranges from 1.2 million to 10 million. The survey was conducted by MetriX Corporation, which is one of the top professional social survey companies in Korea, and the respondents were selected from MetriX's online panels. The data were collected in 2018 using three main types of questions: (1) questions about personal characteristics (demographic and socio-economic), (2) questions about personal attitude and experience of social diversity, and (3) questions about comfortability and willingness to move (WTM) into mixed-income neighborhoods with different levels of income mix. This research got the exemption of the IRB review by the Ulsan National Institute of Science and Technology Institutional Review Board (UNISTIRB-18-36-C), and the informed consent of the survey was made by the participants via the on-line agreement. There is no participation of minors who are under the age of 18. The survey questionnaires translated into English are provided in the S1 Appendix.

To measure comfortability, respondents were asked questions such as "Would you feel comfortable living in the following mixed-income neighborhoods?" To measure WTM, respondents were asked questions such as "Would you be willing to move into the following mixed-income neighborhoods?" Their answers were recorded on a 4-point Likert scale and a binary choice, respectively. The mixed-income neighborhoods were illustrated in pictures, as in Farley et al. [36]. To understand the participants' varying attitudes to different levels of income mix, a matrix consisting of three income groups (low, middle, and high income) and five degrees of the mix was developed by modifying the constructs of Farley et al. [36]. As shown in Fig 1, the five mixed-income levels were: (1) homogenous, (2) slightly mixed, (3) moderately mixed, (4) considerably mixed, and (5) extremely mixed. We intended that the

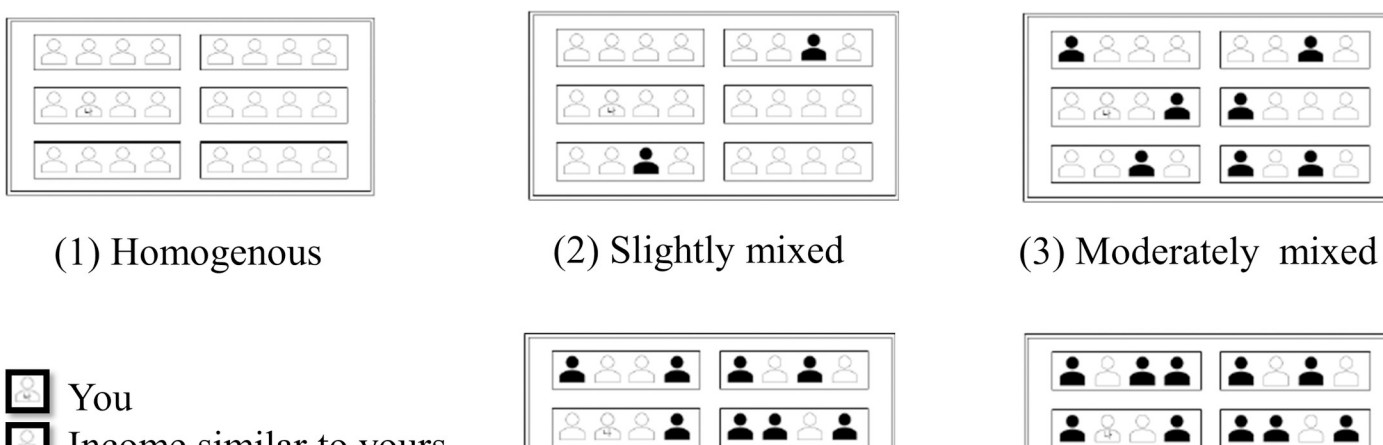

**Fig 1. Descriptions of five different mixed-income neighborhoods.**

respondent is a majority in the slightly mixed and moderately mixed neighborhoods while the respondent is a minority in the extremely mixed neighborhoods.

Regarding personal attitude to social mix, this study measured political propensity, openness to diversity, and community attachment via the survey questionnaire. Political propensity was categorized into progressive, moderately progressive, neutral, moderately conservative, and conservative, and finally classified into three groups in the analysis, namely progressive (combining progressive and moderately progressive), neutral, and conservative (combining conservative and moderately conservative). Openness to diversity was measured using five questions related to respondents' intentions to engage in social interaction with members of a different socioeconomic class. Responses were measured on a 4-point Likert scale. We then calculated the mean value of the answers to all five questions. The first three questions asked about the participants' openness to people from different socioeconomic groups, such as whether they would join their family, befriend them, or become their neighbor. The other two questions inquired about their general acceptance and willingness to live with neighbors from a different socioeconomic background. There were eleven questions about community attachment, asking about topics such as residents' acceptance, belonging, and trust regarding the existing neighborhood. Responses were recorded on a 4-point Likert scale, then the average values were used to represent respondents' community attachment.

Personal experience of social diversity was measured using the survey data and other data sources pertaining to frequency of social interaction with different income groups, neighborhood income diversity, and the share of foreigners in the neighborhood population. Frequency of social interaction was investigated on six different levels of meeting frequency (none, once a year, once in six months, once per month, once per week, and almost every day) with family/relatives, friends and colleagues, and neighbors at different income levels. For instance, if the respondent was classified into the low-income group, their frequency of social interaction with the middle- and higher-income groups was measured. Income diversity index values were calculated at an administrative Dong level using the Simpson index formula, which measures the degree of concentration of individuals among different groups when individuals are classified

**Table 1. Measurements and data sources of variables.**

| Variables | Measurements | Data sources |
|---|---|---|
| **Acceptance of mixed-income communities** | Sum of answers to five questions about the acceptance toward five different levels of mixed-income communities | Survey C1.1–5 |
| **WTM into mixed-income communities** | Sum of answers to five questions about the WTM into five different levels of mixed-income communities (Yes = 1; No = 0) | Survey C2.1–5 |
| **Personal attitude** | | |
| Political propensity | (1) conservative: "conservative" or "moderately conservative"<br>(2) Neutral: neutral<br>(3) Progressive: "progressive" or "moderately progressive" | Survey B1 |
| Openness to diversity | Average values of answers to five questions about respondents' openness | Survey B3.1–5 |
| Community attachment | Average values of answers to eleven questions about respondents' community belonging and their satisfaction | Survey B2.1–11 |
| **Personal experience of social diversity** | | |
| Frequency of interaction | Average values of answers to questions about the frequency of social interactions with other income groups | Survey B4.1.1–2, B4.2.1–2, B4.3.1–2 |
| Income diversity index (Dong level) | Simpson index with six different levels of housing prices at an administrative Dong level | Housing transactions database from MoLIT (2016–2018) |
| Foreigner rate (Dong level) | The ratio between the registered foreigners and the Korean population at an administrative Dong level | Address registration database of Korea |
| **Demographic characteristics** | | |
| Gender | (1) Male, (2) Female | Survey S1 |
| Age | Classified into (1) 20s, (2) 30s, (3) 40s, (4) 50s, (5) 60s+ | Survey S2 |
| Marital status | Classified into (1) single, (2) married, and (3) divorced or widowed (otherwise) | Survey A1 |
| Household size | The number of household members in their current house | Survey A2 |
| School-aged children | Dummy variable whether there are at least one or more children aged under 19 in a household | Survey A3 |
| **Socioeconomic characteristics** | | |
| Housing tenure | Classified into (1) owners, and (2) renters (otherwise) | Survey A5 |
| Educational attainment | Classified into (1) graduate school (when they checked "graduate school graduation"), (2) college (when they checked "undergraduate graduation", or "currently enrolled in graduate school"), and (3) high school or less (otherwise) | Survey A7 |
| Household income | Monthly household gross income (unit: 1,000 KRW) | Survey A8 |
| Neighborhood quality | Median apartment-type housing prices at an administrative Dong level (unit: 1,000 KRW) | Housing transactions database from MoLIT (2018) |
| **Location** | City dummy variables | Survey S3 |

Note: WTM = Willingness to move in; MoLIT = Ministry of Land, Infrastructure and Transport

into the groups [25]. Here, six different levels of housing prices, considering income eligibility for public housing, were used as a proxy for household income levels because household income data are not available on a neighborhood scale in Korea. Last, the foreigner rate was measured using the address registration database as a ratio between the population of registered foreigners and the total Korean population at an administrative Dong level. The higher foreigner rate in a neighborhood the more expose to social diversity in Korea because Korea is relatively homogenous country in terms of race and ethnicity. The operational definition and relevant survey questionnaire, and summary statistics for the main variables used in this study are presented in Tables 1 and 2, respectively.

## Methods of analysis

To analyze residents' comfortability with and WTM into mixed-income communities by income group, we first separated the survey data, based on respondents' self-reported income,

**Table 2. Summary statistics of variables (n = 1996).**

| Variables | | Mean | SD | Min | Max |
|---|---|---|---|---|---|
| Acceptance of mixed-income communities | Middle- and high-income respondents (n = 1229) | 12.56 | 3.13 | 5 | 20 |
| | Low-income respondents (n = 767) | 13.39 | 3.51 | | |
| WTM into mixed-income communities | Middle- and high-income respondents (n = 1229) | 2.53 | 1.66 | 0 | 5 |
| | Low-income respondents (n = 767) | 3.37 | 1.79 | | |
| **Personal attitude** | | | | | |
| Political propensity | Conservative (ref) | 0.2 | - | 0 | 1 |
| | Neutral | 0.51 | | | |
| | Progressive | 0.29 | | | |
| Openness to diversity | | 2.76 | 0.54 | 1 | 4 |
| Community attachment | | 2.48 | 0.54 | 1 | 4 |
| **Personal experience of social diversity** | | | | | |
| Frequency of interaction | With low-income people (n = 1229) | 3.62 | 0.79 | 1 | 6 |
| | With middle-income people (n = 767) | 3.31 | 1.44 | | |
| Income diversity index (Dong level) | | 3.6 | 0.84 | 1.1 | 5.91 |
| Foreigner rate (Dong level) | | 0.04 | 0.07 | 0 | 1.12 |
| **Demographic characteristics** | | | | | |
| Gender | Female (ref.) | 0.45 | - | 0 | 1 |
| | Male | 0.55 | | | |
| Age | 20s (ref.) | 0.21 | - | 0 | 1 |
| | 30s | 0.21 | | | |
| | 40s | 0.23 | | | |
| | 50s | 0.26 | | | |
| | 60+ | 0.09 | | | |
| Marital status | Single (ref.) | 0.34 | - | 0 | 1 |
| | Married | 0.64 | | | |
| | Divorced or widowed | 0.02 | | | |
| Household size | | 3.14 | 1.16 | 1 | 8 |
| School-aged children | Absence of children under 19 (ref.) | 0.64 | - | 0 | 1 |
| | Presence of children under 19 | 0.36 | | | |
| **Socioeconomic characteristics** | | | | | |
| Housing tenure | Owner (ref.) | 0.66 | - | 0 | 1 |
| | Renter | 0.34 | | | |
| Educational attainment | High school or less (ref.) | 0.16 | - | 0 | 1 |
| | College | 0.72 | | | |
| | Graduate school | 0.12 | | | |
| Household income (1,000 KRW) | | 5,529.50 | 7,644.70 | 0 | 150,000 |
| Neighborhood quality (1,000 KRW) | | 341,602 | 263,351 | 42,500 | 2,382,500 |
| **Location** | | | | | |
| | Seoul | 0.44 | - | 0 | 1 |
| | Busan | 0.15 | | | |
| | Daegu | 0.11 | | | |
| | Incheon | 0.13 | | | |
| | Daejeon | 0.06 | | | |
| | Gwangju | 0.06 | | | |
| | Ulsan | 0.05 | | | |

Note: WTM = Willingness to move; SD = Standard Deviation; 1 US $ is equivalent to 1,100 KRW.

into a middle- and high-income group and a low-income group (middle- and high-income respondents were combined due to the relatively small number of participants that reported themselves as "a high-income household"). Respondents' comfortability with and WTM into mixed-income communities with different levels of mix are presented for two cases on a graph showing low-income respondents' acceptance (from low to middle/high) and higher-income respondents' acceptance (from middle/high to low).

Then, an ordinal logistic regression model for each case was estimated to examine the factors affecting the respondents' comfortability and WTM. Since the different levels of income mix are ordinal rather than on an interval scale, the ordinal logistic regression model is an appropriate method of analysis for our study. In conducting the ordinal logistic regression, we decided to reduce the five levels of mix to three main levels, namely (i) a "homogeneous with you" neighborhood, (ii) a "you are in the majority" neighborhood (called majority and formed by combining slightly and moderately mixed neighborhoods), and (iii) a "you are in the minority" neighborhood (called minority and formed by combining considerably and extremely mixed neighborhoods). This regression model, with three levels of income mix, provides more concise and comparable results for the types of mixed-income communities. Indeed, the estimated results of the ordinal logit model using these three levels of income mix are similar to those obtained using the five levels of income mix. For each income class, eight models were generated, with four models dedicated to acceptance and willingness. One model evaluates general acceptance (average points from all five different levels of mixed-income communities), and three models use three levels of income mix. The values of the dependent variables in each model were calculated to determine these components' weights, such as the value of total acceptance equal to the sum of the point from all levels of income mix.

## Results and discussion

After reviewing the data, responses from 1,996 respondents were included in the analysis because there is no available neighborhood quality (median housing price) information for four responses. Respondents were then classified into middle- and high-income residents ($n = 1,229$), and low-income residents ($n = 767$) based on their self-reported income.

### Perception of mixed-income communities by levels of mix

Fig 2 shows participants' perceptions of different levels of income mix, based on the understanding that low-income residents would mainly be mixed into middle-income neighborhoods and middle- and high-income residents would mainly be mixed into low-income neighborhoods. This figure illustrates a downward trend in both participant comfortability with and WTM into mixed-income neighborhoods along with the progression from homogeneous to extremely mixed communities. It is indicated that homogeneous neighborhoods reached the highest proportion across the five levels of mix, which supports the argument that people prefer to live among others who have similar incomes.

In a comparison between the two self-reported income groups, there is a 3%–4% difference in their comfortability with and WTM into homogenous neighborhoods. A higher degree of uncomfortableness with low-income group members could be interpreted as attributable to higher tension levels experienced in the dense, highly-concentrated poverty context [3]. About 20% of the people in the two income groups do not have a positive perception of homogeneous neighborhoods; hence, the income group they would like to see in their neighborhood is presented in charts showing the other four levels of mix. Compared to the middle-/high-income group, low-income people's comfortability with and WTM into mixed-income communities are higher; however, there is a remarkable decrease in these values following an increase in the

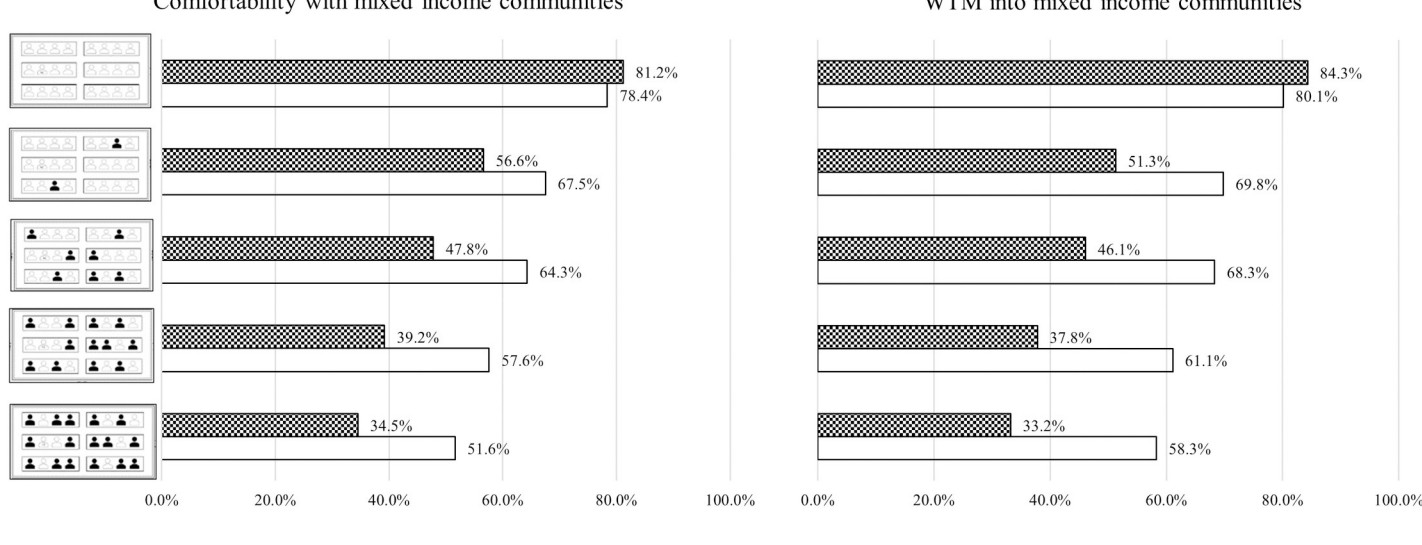

**Fig 2. Comfortability with and WTM into mixed-income neighborhoods.**

level of income mix. This trend reveals that low-income people are more likely to accept mixed-income communities than higher-income individuals.

Regarding levels of mix involving low-income people, more than 50% of the respondents reported that they feel comfortable with considerable and extreme levels of mix. However, more than 50% of the higher-income group indicated that they would feel comfortable only in a slightly mixed community. Overall, for each level of income mix, the share of higher-income respondents who indicated feeling comfortable was about 10%~25% lower than the share of low-income group members. These findings indicate that the higher-income group could mix with low-income people in a slightly mixed community, whereas low-income people are more willing to accept higher degrees of mix in the context of the mixed-income policy. Based on these results, this study compresses the five levels of mix into three degrees of mix to more easily capture the change in acceptance, which is shown in the third subsection in Results and Findings. Higher-income respondents' WTM showed a similar pattern as their comfortability, but the share that would be accepting is relatively lower than the share that indicated feeling comfortable. This is because compared to comfortability, WTM more strongly reflects respondents' intentions regarding mixed-income communities.

### The effects of personal attitude and experience on general acceptance of mixed-income communities

Based on the results of the ordinal logistic model, the scores for total general comfortability with and WTM into mixed-income communities are shown in Table 3. The results showed that openness to diversity and interaction frequency with other groups had a significant effect on respondents' perception. While political propensity and community attachment only impacted higher-income individuals, the income diversity index could increase low-income respondents' willingness to accept mixed-income communities.

**Personal attitude.** Independent variables related to personal attitude are political propensity, openness to diversity, and community attachment. The estimated results in Table 3 show that progressive middle- and high-income people have 41% higher acceptance of feeling

**Table 3. Results of ordinal logistic regression (odds ratio).**

| Variables | Middle- and high-income to mixed low-income neighborhood | | Low-income to mixed middle-income neighborhood | |
|---|---|---|---|---|
| | Acceptance | Willingness to move into | Acceptance | Willingness to move into |
| **Personal attitude** | | | | |
| *Political propensity (ref. conservative)* | | | | |
| Neutral | 1.17 | 0.78 | 0.83 | 0.87 |
| Progressive | 1.41* | 0.93 | 1.32 | 1.29 |
| *Openness to diversity* | 4.13*** | 2.74*** | 3.81*** | 2.94*** |
| *Community attachment* | 0.82* | 0.89 | 0.85 | 0.93 |
| **Personal experience of social diversity** | | | | |
| *Frequency of interaction:* | | | | |
| - with low-income people | 0.70*** | 0.74*** | | |
| - with middle-income people | | | 1.27*** | 1.24*** |
| *Income diversity index* | 0.94 | 0.94 | 1.14 | 1.20* |
| *Foreigner rate* | 1.47 | 1.24 | 2.47 | 4.48 |
| **Control variables** | | | | |
| *Gender (ref. female)* | | | | |
| Male | 1.12 | 1.03 | 0.76 | 0.77 |
| *Age (ref. 20s)* | | | | |
| 30s | 0.90 | 0.87 | 0.85 | 1.16 |
| 40s | 0.53*** | 0.65* | 0.88 | 1.29 |
| 50s | 0.60** | 0.67* | 1.05 | 1.54 |
| 60s | 0.49** | 0.57* | 0.85 | 1.43 |
| *Marital status (ref. single)* | | | | |
| Divorced and widowed | 0.37* | 0.49 | 0.82 | 0.78 |
| Married | 0.74 | 0.75 | 1.13 | 0.89 |
| *Household size* | 1.00 | 1.02 | 1.20** | 1.12 |
| *School-aged children (ref. absence of children under 19)* | | | | |
| Presence of children under 19 | 1.40* | 1.32 | 0.87 | 0.84 |
| *Housing tenure (ref. owner)* Renter | 1.00 | 1.30* | 1.09 | 0.98 |
| *Educational attainment (ref. high school or less)* | | | | |
| College | 0.75 | 0.83 | 1.03 | 1.27 |
| Graduate school | 0.70 | 0.63* | 1.00 | 1.13 |
| *Household income* | 0.93 | 0.90 | 0.94 | 0.87* |
| *Neighborhood quality* | 0.98 | 0.92 | 0.83* | 0.89 |
| *Location (ref. Seoul)* | | | | |
| Busan | 1.21 | 1.04 | 0.61* | 0.56* |
| Daegu | 0.97 | 0.95 | 1.21 | 1.44 |
| Incheon | 1.44* | 1.43 | 0.82 | 0.80 |
| Daejeon | 0.80 | 0.92 | 0.83 | 0.74 |
| Gwangju | 0.90 | 0.94 | 0.79 | 0.96 |
| Ulsan | 1.03 | 1.07 | 0.51* | 0.87 |
| **Number of observations** | 1229 | | 767 | |
| **Cragg and Uhler's pseudo *R*-squared** | **0.212** | **0.148** | **0.238** | **0.174** |

***p < 0.001

**p < 0.01

*p < 0.05

comfortable in a mixed-income community compared to those whose political propensity is conservative consistent with previous studies [27, 35]. Here, the ideological pattern refers to the progressiveness of thoughts related to race and income class distinctions, so residents who subjectively identify as "conservative" lean toward conformity, tradition, and security [33]. For this reason, progressive inhabitants demonstrate stronger acceptance of mixed-income communities. Political propensity strongly affects residents' perception of mixed-income communities and thus needs to be reviewed as a prominent element.

It was also found that openness to diversity plays an important role. Consistent with Sawyerr et al. [33], the results indicated that both low- and higher-income residents who were more open-minded demonstrated 3.8 and 4.1 times the odds of having more comfortability with mixed-income communities compared to those with low openness, respectively. A strong social network is strongly associated with a socially-mixed context [37]. It follows that if people are more open to broadening their social networks in a mixed-income society, their attitude toward a diverse neighborhood will be more favorable. However, it is not accurate to state that those with a strong social network or considerable social capital are automatically more motivated to live in a mixed-income neighborhood; this is so because quantity, in addition to network diversity, is important [45]. For instance, a strong social network that incorporates a large number of family members, relatives, and close friends within mainly one neighborhood could evidence an isolationist or homogeneous lifestyle. Openness to diversity indicates a tolerant attitude toward a diversity of inhabitants, regardless of their income, location, and social position.

Compared to acceptability, the odds ratio indicates that someone who has a high degree of openness to diversity will also reveal a remarkable difference in their WTM into a mixed-income neighborhood. In this study's lower- and higher-income groups, the residents who demonstrated greater openness were, respectively, 2.9 and 2.7 times the odds of having WTM into a mixed-income neighborhood compared to those who demonstrated a less open-minded attitude. However, there is a considerable difference in the estimated odd ratios between acceptability and WTM. This difference between the odds ratios of comfortability and WTM indicates that even if respondents have a positive attitude toward a mixed-income neighborhood or social diversity, they must also perceive other perspectives when deciding whether to move in, which could include security concerns, inequality, and physical and social detachment [46].

Community attachment refers to residents' experience in their current neighborhood in terms of acceptance, trust, and belonging. The results showed that respondents who had high scores for attachment to their existing neighborhood tended to have higher acceptance of feeling comfortable in mixed-income communities than those who were detached from their community. Those who were satisfied with their current neighborhoods indicated a desire to continue living there and would be less likely to consider moving to another neighborhood [19, 44]. Residents' positive experiences with regard to their satisfaction with the quality of their physical surroundings would decrease their willingness to move to other local neighborhoods.

**Personal experience of social diversity.** Experience with social diversity consists of the frequency of social interaction with different income groups, the income diversity index, and the number of foreigners living in the neighborhood (the foreigner rate). Interestingly, the frequency of interaction with different income groups showed opposite effects between the low- and higher-income groups. On the one hand, higher-income respondents' frequent meetings with low-income people negatively affected their comfortability with and WTM into mixed-income communities. A one-unit increase in the frequency of interactions with low income groups led to 30% and 26% lower comfortability and WTM among higher-income

respondents, respectively. Considering the finding of Raynor et al. [12] that positive social interaction experiences with social housing residents could reduce the stigmatization of social housing, the social interactions between higher-income and lower-income residents might not be positive in Korean cities. Middle- and high-income residents who interact frequently with the lower-income class could perceive the negative consequences of a mixed neighborhood (i.e., security concerns), which would explain their lower WTM [46, 47]. On the other hand, low-income respondents with higher interaction frequency with the higher-income group were more likely to accept mixed-income communities. A one-unit increase in the frequency of interactions with higher- income groups led to 30% higher acceptance of feeling comfortable in a mixed-income community among lower-income respondents. Low-income people might feel that a mixed-income neighborhood is not only a good opportunity to access better facilities but also to enhance their chances of finding a good job and reduce inequality [19, 48]. Generally, residential acceptance is strongly impacted by resident choice factors and can be explained by residential expectations and satisfaction [49].

The Simpson index indicates that income-diverse environments in local neighborhoods could influence residents' social interaction opportunities. The income diversity index showed that if one score index increases, low-income residents' WTM also increases by approximately 20%. However, in other cases, there is no statistically significant effect. A community with greater income diversity results in a higher degree of socialization characterized by urban diversity and affects respondents' experiences [46]. The less statistically significant results of the income diversity index maybe because the income diverse condition itself does not ensure an increase in positive experiences among different social groups.

Finally, the foreigner rate was not found to be a significant variable impacting residents' perception of a mixed-income society, which might be due to the lack of a relationship between culture and income diversity. Moreover, a higher foreigner rate does not automatically guarantee more social interaction and communication among different social groups.

**Controlled variables.** For demographic variables, this study asserts that marital status could an important factor in determining an individual's attitude toward mixed-income communities in Korea. People who are divorced and widowed revealed significantly less comfortability with the idea of a mixed-income neighborhood; their level of comfort was 63% lower than that of single individuals, which is a meaningful difference. However, the probability of this variable accounted for only 2% of the sample (see Table 2); thus, future studies must examine the effect of marital status on resident perception. Regarding the socio-economic component, housing tenure only affects WTM within the higher-income group. Low-income renters could gain occupational opportunities and access to better amenities in a mixed-income community, but owners may also opt to stay if they perceive property in that neighborhood to be a good investment [47, 48]; therefore, this study revealed that no difference exists by housing tenure in terms of preference for mixed-income communities among low-income people in Korea.

## The effect of personal attitude and experience on acceptance of mixed-income communities at different levels of mix

To clarify that change in acceptance depends on personal attitude and experience of social diversity across different levels of income mix, we present changes in the estimated coefficients between the homogenous (no-mix neighborhood), the majority (slightly and moderately mixed neighborhoods), and the minority classifications (considerably and extremely mixed neighborhoods) in Fig 3. Overall, the patterns are similar to the estimated results of the ordinal logit model for general comfortability and WTM.

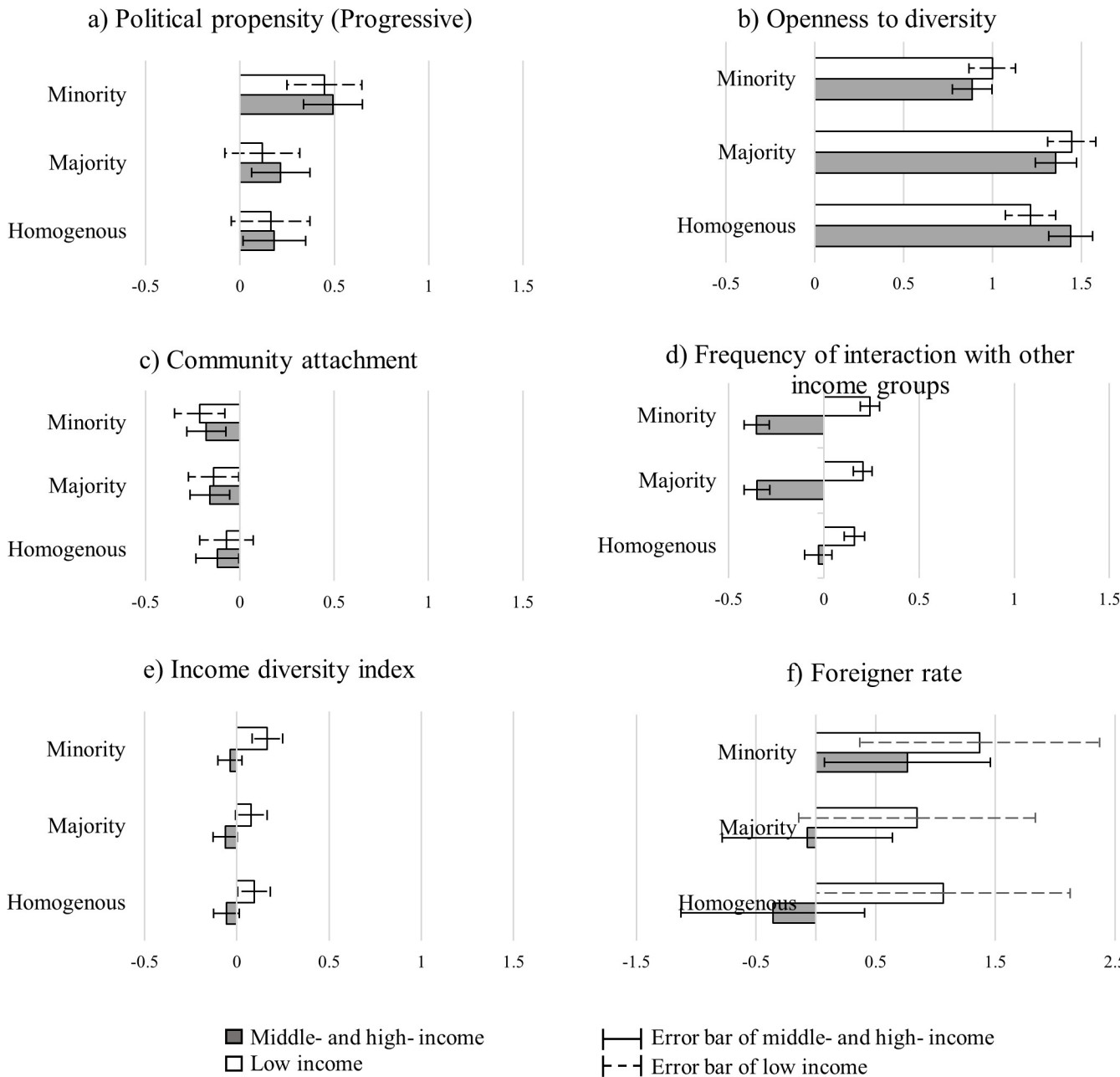

**Fig 3. The change in coefficients in the ordered logistic regression.**

The differences between the estimated coefficients for each level of mix are compared using the size and directions of the estimated coefficients and their standard errors. If there is no overlap in the standard error intervals between two successive levels of income mix, there may be a difference in residents' acceptance by the level of income mix. For instance, residents' preferences by political propensity (progressive versus conservative) and community attachment do not show a big difference between different income levels, as shown in Fig 3A and 3C, because there is a lot of overlap in the coefficient estimation, considering the standard errors. However, personal openness to diversity and frequency of interaction were revealed to

be meaningful factors to change respondents' preferences across different levels of income mix (Fig 3B and 3D).

In terms of personal openness to diversity (Fig 3B), middle- and high-income participants showed a significantly lower preference for the minority situation than the majority one, while no meaningful difference was found between the homogenous and majority classifications. Although the openness of those in the higher-income category helps to raise their acceptance of mixed-income communities, this trend revealed that they would prefer to live in either homogenous communities or those in which the majority of residents in their neighborhood are classified as belonging to the higher-income class. In contrast, low-income respondents showed an increased tendency to prefer the majority situation over the homogenous one, whereas preference for the minority situation was considerably lower than that for the majority. This change proved that those in the low-income category were more likely to choose mixed-income communities than those classified as homogenous, except for when the low-income resident percentage is lower than other income groups.

Regarding the perspectives on interaction, if middle- and high-income residents connect frequently with low-income people, their acceptance of mixed-income communities decreases significantly as the level of mix increases. Conversely, although the result is not significant, low-income respondents demonstrated a trend toward greater acceptance when the frequency of interaction with middle-income people increased.

## Conclusions

This study explores perceptions of and preferences for mixed-income communities in Korea. Low-income people show higher acceptance (over 50% comfortability and WTM) if they can move to a mixed neighborhood with middle-income households, even at the considerably and extremely mixed levels. However, the higher-income group would prefer a slight mix. This mismatch presents a significant challenge for urban planners in preparing mixed-income policies.

Besides the impact of socioeconomic and demographic factors on the acceptability of mixed-income communities within each income group, political propensity and frequency of interaction with other social groups are two strongly significant determinants. More open-minded participants demonstrated a higher acceptance of mixed-income neighborhoods. For social interaction, higher levels of interaction with low-income residents would decrease WTM among higher-income respondents, while a stronger connection with middle-income people could raise the level of acceptability among those in the low-income category.

These findings could support policymakers and urban planners as they try to promote socially-mixed developments while grappling with divergent preferences. First, although the level of acceptance can decrease with an increase in the level of mix, there are open-minded and politically progressive people who could be potential residents of mixed-income communities. Second, the interaction between low-income and higher-income inhabitants could be considered evidence of growth in their willingness to accept the idea of moving into mixed-income communities, in particular, among low-income residents. On the other hand, institutional organizations and routine activities should aim to highlight the positive aspects of the low-income lifestyle to enhance higher-income people's image of low-income individuals [3]. Finally, since residents' social interaction and neighborhood attachment could increase with their experience of good conditions, the design of mixed-income neighborhood facilities and amenities plays an important role in social inclusion developments.

Additionally, this study, with a Korean case study, contributes to the social mix studies literature and could apply to other racially homogenous countries such as Japan and Vietnam.

This study also found evidence of the effects of personal attitude and experience with social diversity on residents' perception of mixed-income communities in their cognitive-behavior circulation. For example, social attitude and interactive frequency exerted strongly significant impacts. Further studies could refer to this framework to focus not only on demographic and socioeconomic factors, but also on perceptions and prior experiences, all of which should be considered systematically in the context of resident behavior and bottom-up policy perspectives.

The gap between resident perception and real behavior must be bridged in future studies. In reality, residents' response depends not only on their attitude toward mixed-income neighborhoods, but also on structural elements of the external environment, such as neighborhood design and public services, which contribute to enhanced social cohesion and control [4]. The connections between people's perception and their actual behavior in terms of residential location choices for mixed-income communities could be systematically explored.

## Supporting information

**S1 Appendix. The survey on residents' perception and acceptance towards shared housing and mixed-income communities.**
(DOCX)

**S1 Data. The raw data of survey questionaries and the data sets used for the analysis.**
(XLSX)

## Author Contributions

**Data curation:** Duc Trung Luu.

**Formal analysis:** Duc Trung Luu.

**Investigation:** Dao Chi Vo.

**Methodology:** Dao Chi Vo, Jeongseob Kim.

**Project administration:** Jeongseob Kim.

**Supervision:** Jeongseob Kim.

**Validation:** Duc Trung Luu, Dao Chi Vo.

**Visualization:** Duc Trung Luu.

**Writing – original draft:** Duc Trung Luu.

**Writing – review & editing:** Dao Chi Vo, Jeongseob Kim.

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
