## [Decision Letter · Decision Letter 0]

27 Jan 2021

PONE-D-20-38542

Personal attitude or experience? Which factors influence residents’ acceptance of mixed-income communities?

PLOS ONE

Dear Dr. Kim,

Thank you for submitting your manuscript to PLOS ONE. After careful consideration, we feel that it has merit but does not fully meet PLOS ONE’s publication criteria as it currently stands. Therefore, we invite you to submit a revised version of the manuscript that addresses the points raised during the review process.

We look forward to receiving your revised manuscript.

Kind regards,

Andrew T. Carswell

Academic Editor

PLOS ONE

Journal Requirements:

2. Please provide a copy of the questionnaire used in the study, as a supplemental file.

'This work was supported by the National Research Foundation of Korea (NRF) Grant funded by the Korean Government(MSIP) (No.NRF-2015R1A5A7037825).'

'The funders had no role in study design, data collection and analysis, decision to publish, or preparation of the manuscript.'

Reviewers' comments:

Reviewer's Responses to Questions

**Comments to the Author**

1. Is the manuscript technically sound, and do the data support the conclusions?

Reviewer #1: Yes

Reviewer #2: Yes

2. Has the statistical analysis been performed appropriately and rigorously? 

Reviewer #1: Yes

Reviewer #2: Yes

3. Have the authors made all data underlying the findings in their manuscript fully available?

Reviewer #1: Yes

Reviewer #2: No

4. Is the manuscript presented in an intelligible fashion and written in standard English?

Reviewer #1: Yes

Reviewer #2: Yes

5. Review Comments to the Author

Reviewer #1: Well-written; introduction lays out a strong argument of why this study is important. Unique perspective to focus on the consumer in regards to mixed-income communities.

p.3 1st paragraph: “Yonker” should be “Yonkers”

p.5 last paragraph, “as seen” should be added before “in some stigmatized French neighborhoods”

p.6 “The processes of economic reform and institutional transition under neo-liberalism have affected residents’ ideology and value judgment, leading to a change in residents’ preferences and support for inclusive housing policies in Chinese cities [35]. Accordingly, more residents have come to accept economic individualism, which is possibly an emerging challenge for social mix [35].” – this seems contradictory?

What was response rate of survey? If low, this should be noted as a limitation

Helpful to explain the similarities and differences of the 7 metropolitan areas chosen and why these 7 were chosen. Any city differences in results?

An appendix with a copy of the survey would be helpful for readers to review.

Table 1: Political Propensity shows max at 1, but this variable shows progressive as being 2. That would lead me to believe that max is 2, not 1?

Top of page 10: “wights” should be “weights”

Top of page 11: why were 10 responses excluded?

Table 1, Figure, 2, Table 2, and Figure 3 should be placed after the first paragraph where it is referenced, not a few paragraphs below.

Results seem to confirm much of the existing literature and this should be noted in the discussion as well as what existing literature it confirms.

Reviewer #2: I did not see a data set that you had submitted, but I might have missed it. Please provide that.

On page 2, your reference to “the actual field of communities” seems a little awkward. Substituting a phrase in there such as “urban planning” or “community development” might be less awkward.

On page 3, I am wondering if the “Yonker” program should actually be “Yonkers”.

On page 6, you mention “social contact theory” and am wondering if you might have meant “social contract theory”.

In the survey sampling design, it is not clear how the populations were chosen. Was there a database of mixed income properties? How many properties are we talking about here?

In Figure 1, is this your own way of operationalizing degree of mix? Panels 4 and 5 look pretty much the same to me.

I like that you used the income diversity index…this seems like the proper way to determine some form of mixed. Having said that, I think adding a sentence or two about the Simpson index methodology might be appropriate here as well.

On page 9, you mentioned a foreigner rate in an address registration data base. I am not familiar with this concept. Also, if the focus is on degree of mix by income level, what does foreign-born have to do with anything?

Also, if you could convert the currency into dollars, that would be helpful to US readers to understand perspective.

It might not be a bad idea to include the survey instrument as an appendix. It seems like the manuscript has enough room for it to be included.

On page 13, I do not think that the interpretation of the coefficient for community attachment is correct. Instead of .81 times lower acceptance, it is technically .81 times of average acceptance for those with low scores for community attachment. Perhaps it is better to say 19% lower?

You make the same interpretation issues with the other variables as well. For example, with the statement “Lower-income individuals who often meet higher-income people demonstrated about 1.3 times higher acceptance than those who rarely interact with higher-income people”, it should actually be “30% higher acceptance”. The way it reads now, it seems like it almost be 130% higher. Please go through all of these coefficient interpretations.

6. PLOS authors have the option to publish the peer review history of their article (what does this mean?). If published, this will include your full peer review and any attached files.

Reviewer #1: No

Reviewer #2: No

---

## [Author Response · Author response to Decision Letter 0]

11 Mar 2021

We very much appreciate the thoughtful recommendations and comments of the reviewers and editor. Based on the feedback, we made revisions by clarifying the explanations and interpretation of the results. Also, we provided the survey questionnaires in Appendix as supporting information, and the correspondence between survey questionaries and variables is presented in Table 1. Our responses to the reviewers are provided in a separate file attched.

---

## [Decision Letter · Decision Letter 1]

8 Apr 2021

Personal attitude or experience? Which factors influence residents’ acceptance of mixed-income communities?

PONE-D-20-38542R1

Dear Dr. Kim,

We’re pleased to inform you that your manuscript has been judged scientifically suitable for publication and will be formally accepted for publication once it meets all outstanding technical requirements.

Kind regards,

Andrew T. Carswell

Academic Editor

PLOS ONE

Additional Editor Comments (optional):

Reviewers' comments:

Reviewer's Responses to Questions

**Comments to the Author**

1. If the authors have adequately addressed your comments raised in a previous round of review and you feel that this manuscript is now acceptable for publication, you may indicate that here to bypass the “Comments to the Author” section, enter your conflict of interest statement in the “Confidential to Editor” section, and submit your "Accept" recommendation.

Reviewer #1: All comments have been addressed

Reviewer #2: All comments have been addressed

2. Is the manuscript technically sound, and do the data support the conclusions?

Reviewer #1: Yes

Reviewer #2: Yes

3. Has the statistical analysis been performed appropriately and rigorously? 

Reviewer #1: Yes

Reviewer #2: Yes

4. Have the authors made all data underlying the findings in their manuscript fully available?

Reviewer #1: Yes

Reviewer #2: Yes

5. Is the manuscript presented in an intelligible fashion and written in standard English?

Reviewer #1: Yes

Reviewer #2: Yes

6. Review Comments to the Author

Reviewer #1: (No Response)

Reviewer #2: The authors have provided a good paper worthy of publication, and seem to hve addressed all of my initial concerns.

7. PLOS authors have the option to publish the peer review history of their article (what does this mean?). If published, this will include your full peer review and any attached files.

Reviewer #1: No

Reviewer #2: No

---

## [Editor Report · Acceptance letter]

13 Apr 2021

PONE-D-20-38542R1 

Personal attitude or experience? Which factors influence residents’ acceptance of mixed-income communities? 

Dear Dr. Kim:

I'm pleased to inform you that your manuscript has been deemed suitable for publication in PLOS ONE. Congratulations! Your manuscript is now with our production department. 

Kind regards, 

on behalf of

Dr. Andrew T. Carswell 

Academic Editor

PLOS ONE